# WE economy: Potential of mutual aid distribution based on moral responsibility and risk vulnerability

**Takeshi Kato** [ID] *

Hitachi Kyoto University Laboratory, Open Innovation Institute, Kyoto University, Kyoto, Japan

* kato.takeshi.3u@kyoto-u.ac.jp

## Abstract

Reducing wealth inequality is a global challenge that requires the transformation of the economic systems that produce inequality. The economic system comprises: (1) gifts and reciprocity, (2) power and redistribution, (3) market exchange, and (4) mutual aid without reciprocal obligations. Current inequality stems from a capitalist economy consisting of (2) and (3). To sublimate (1), the human economy, to (4), the concept of a "mixbiotic society" has been proposed in the philosophical realm. In this society, free and diverse individuals mix, recognize their respective "fundamental incapability," and sublimate them into "WE" solidarity. Moreover, the economy must have a moral responsibility as a co-adventurer and consider its vulnerability to risk. This study focuses on two factors of mind perception—moral responsibility and risk vulnerability—and proposes a novel wealth distribution model between the two agents following an econophysical approach, whereas the conventional model dealt with redistribution through taxes and institutions. Three models are developed: a joint-venture model in which profit/losses are distributed based on their factors, a redistribution model in which wealth stocks are redistributed periodically based on their factors in the joint-venture model, and a "WE economy" model in which profit/losses are distributed based on the ratio of each other's factors. A simulation comparison reveals that WE economies are effective in reducing inequality, resilient in normalizing wealth distribution as advantages, and susceptible to free riders as disadvantages. However, this disadvantage can be compensated for by fostering fellowship and using joint ventures. This study presents the effectiveness of moral responsibility and risk vulnerability, complementarity between the WE economy and joint economy, and the direction of the economy in reducing inequality. Future challenges include developing an advanced model based on real economic analysis and economic psychology and promoting its fieldwork for worker coops and platform cooperatives to realize a desirable mixbiotic society.

## Introduction

Wealth inequalities and disparities are major social issues worldwide. According to the World Inequality Report 2022, the top 1% of the richest people account for 38% of the world's wealth

**Data Availability Statement:** All relevant data are within the manuscript and its Supporting Information files.

**Funding:** This work was supported by Japan Society for the Promotion of Science Topic-Setting

Program to Advance Cutting-Edge Humanities and Social Sciences Research Grant Number JPJS00122679495. The funder had no role in study design, data collection and analysis, decision to publish, or preparation of the manuscript.

**Competing interests:** The author has declared that no competing interests exist.

[1], and according to the World Economic Forum, only eight men at the top have the same wealth as the poorest 3.6 billion people [2]. According to the 2022 report, the global Gini index reached 0.7 [1], well above the warning level of 0.4 for social unrest [3]. Solving the problem of inequality is an urgent issue because social unrest creates a vicious cycle that lowers productivity, increases inequality, and fuels social unrest [4].

The United Nations Sustainable Development Goals include Goal 10, which calls for reducing inequality; promoting social, economic, and political inclusion; and developing fiscal and social policies that promote equality [5]. Goals 1, 2, 8, and 16 call for eradicating poverty and achieving zero hunger, inclusive economic growth, and fair and inclusive institutions, respectively [5]. At the World Economic Forum 2017, a gathering of political and business leaders, reducing inequality was on the main agenda [6]. To achieve these goals, it is essential to formulate policies and institutions based on economic relationships that produce inequality—that is, the mode of wealth exchange.

Polanyi, an economist, identifies three modes of economic relations: (1) reciprocity, (2) redistribution, and (3) market exchange [7]. Graeber, an anthropologist, presents (2') hierarchy, (3') exchange, and (4') foundational communism as three moral principles involved in economic relations [8], while Karatani, a philosopher, presents four modes of exchange: (1") reciprocity, (2") plunder and redistribution, (3") commodity exchange, and (4") advanced recovery of reciprocity [9]. Of these, (1) and (1') present a human gift economy with an obligation of return; (2), (2'), and (2") present a power economy with tax collection and redistribution; (3), (3'), and (3") present a market economy with a non-human exchange of goods and money; and (4') and (4") present a human mutual aid economy without an obligation of return, that is sublimated from a gift economy. The capitalist economy that produces the current inequality is a combination of power and market economies [8, 9]. Graeber and Karatani advocate a transformation to a (4') and (4") human economy as a prescription for the inequality problem. The Islamic economy is a combination of redistribution based on the morals of the Islamic code instead of power (*waqf*, *sadaqah*, and *zakat*) and a joint economy that prohibits interest (*mudaraba*, *murabaha*, and *salam*) [10, 11]; it is a capitalist economic alternative and a possible stepping-stone to a human economy [8].

Philosopher Deguchi introduces the concept of a "mixbiotic society," a further development of the symbiotic society, as a social vision corresponding to (4') and (4") above [12]. This is a society in which free and diverse individuals, "I," mix, recognize their respective "fundamental incapabilities," and sublimate them into "WE" solidarity [12, 13]. In a mixbiotic society, individuals entrust each other and cooperate in a "WE" community. Regarding the human economy in a mixbiotic society, the "WE economy" in this study, Deguchi states in his book [14]:

- Good WE: Fellowship, equality, hollowness without a power center, cooperativity, voluntary participation, and softened WE.

- Bad WE: Totalitarianism, exclusivism toward the outside, peer pressure toward the inside, and hardened WE.

- Good WE are co-adventurers who participate while accepting the risk together.

- WE members qualify as co-adventurers (risk-takers) because they are vulnerable and frangible.

- Co-adventurers in the same boat may have economic class divisions, but they are on equal footing as communities of destiny.

- Co-adventurers are weighted according to their moral responsibility, but weighting is only a quantitative differentiation between the received profit and associated risk (latch).

- The risk-return allocation is increased or decreased but not monopolized, and everyone receives a return for the risk taken.

To summarize Deguchi's discourse, two things are important in the "WE Economy": moral responsibility and risk vulnerability. Here, the dimensions of mind perception are informative. According to psychologists, Gray and Wegner, mind perception can be divided into two main dimensions: agency (capacities such as self-control, morality, and memory) and experience (capacities such as hunger, fear, and pain) [15]. As shown in S1 Fig, characters including a fetus, baby, girl, adult man, adult woman, a man in a persistent vegetative state, and a dead woman were mapped on a two-dimensional plane. Agency corresponds to a moral agent (giver of moral action: moral responsibility) and experience of moral patients (recipient of pain: risk vulnerability) [16, 17]. Based on this, the WE economy makes co-adventurers undertake risk (latch) by their moral responsibility and allocate returns in consideration of their responsibility and risk vulnerability. Considering risk vulnerability and moral responsibility involves prioritizing future generations and the socially vulnerable. Integrating both aspects can foster a more humane, mutual-aid economy.

Before examining the possibility of the WE economy in reducing wealth inequality, I consider the leading review literature in economics on wealth inequality. Concerning the heterogeneity of economic agents such as inequality, there are known efforts to introduce microeconomic heterogeneity into macroeconomics [18]. There, the main focus is on the endogenous heterogeneity of individuals and their insurance against exogenous risks, with particular attention to time preference as a heterogeneity [19]. However, moral responsibility and risk vulnerability have never been addressed as heterogeneity. An approach that introduces microeconomic interactions into macroeconomics is the agent-based model (ABM) [20], although it targets firms, governments, banks, and households to maximize profits. Additionally, morality has not been addressed.

In microeconomics, econophysics is an approach that uses ABM and focuses specifically on wealth distribution and inequality. According to reviews in this field, various wealth distribution models have been proposed based on the analogy of the kinetic energy exchange of gaseous particles and other dynamics, and wealth distributions and inequalities, such as exponential, power, gamma, and delta distributions, have been considered [21, 22]. Recent studies have modeled income and inheritance taxes [23], social class and inheritance [24], tax exemptions for the poor [25], contributions of surplus stock [26], interest businesses, joint ventures and redistribution [27], and redistribution and mutual aid [28]. However, none of these models have addressed moral responsibility and risk vulnerability.

I have referred to the review literature on macroeconomics, microeconomics, and econophysics, and discovered a notable research gap with the WE economy in that existing models do not address moral responsibility and risk vulnerability. However, the econophysics approach is more consistent with the abovementioned (2), (2'), (2") tax collection and redistribution, (3), (3'), (3") market exchange, (4') and (4") mutual aid. Among these econophysical models, it is appropriate to refer to the joint-venture model to compare with the WE economy as co-adventurers, as they are the most similar in terms of wealth distribution.

Therefore, this study aims to fill the research gap between the existing literature and the WE economy based on moral responsibility and risk vulnerability and to present the potential and applicability of the WE economy in resolving wealth inequality. I formulate a new mathematical model of the WE economy based on moral responsibility and risk vulnerability by referring to the above joint-venture model [27] and simulating wealth distribution and inequality. By comparing joint ventures and WE economies, I aim to demonstrate the model's effectiveness, which encapsulates moral responsibility and risk vulnerability, and gain insights into the WE economy from the capitalist economy that generates inequality.

The remainder of this paper is organized as follows. In the Methods section, the conventional joint-venture model is introduced and a redistribution model based on moral responsibility and risk vulnerability in the joint-venture model as well as a distribution model based on responsibility and vulnerability in the WE economy are presented. The Results section presents the simulation results for wealth distribution and the Gini index of inequality and refers to historical surveys to validate the WE economy model. Based on these results, I then discuss the real-world applicability of the joint ventures with redistribution and WE economies as alternatives to capitalist economies in the Discussion section. Finally, the future challenges including research issues and empirical fieldwork are presented.

## Methods

### Moral responsibility and risk vulnerability

First, using the agency and experience scores of Gray and Wegner shown in S1 Fig [15], values for moral responsibility and risk vulnerability are established. Excluding a fetus, a man in a persistent vegetative state, and a dead woman, who are not involved in economic activities, a baby, girl, adult man, and adult woman ride on a straight line on a two-dimensional plane. From baby to adult, agency, which corresponds to moral responsibility, roughly changes from 0.2 to 1.0, and experience, which corresponds to risk vulnerability, roughly changes from 1 to 0.8. The age-specific population distribution includes a stationary type with a constant population for each age group, an expansive type with a large population of young people (population growth), and a constrictive type with a small population of young people (population decline) [29]. Here, I assume the stationary type, not the expansive type, with a short life expectancy, nor the constrictive type, with extreme aging. That is, assuming an even distribution of the number of people from babies to adults, the moral responsibility, $\rho_{Mi}$, and risk vulnerability, $\rho_{Ri}$, of the $i$-th agent ($i = 1,2,\cdots,N$) among $N$ agents can be expressed as Eq (1) where $\rho_i$ is the product of $\rho_{Mi}$ and $\rho_{Ri}$.

$$\rho_{Mi} = 0.2 + \frac{0.8}{N} \cdot i,$$

$$\rho_{Ri} = 1 - \frac{0.2}{N} \cdot i,$$

$$\rho_i = \rho_{Mi} \cdot \rho_{Ri}. \tag{1}$$

### Joint-venture and redistribution models

As the econophysical model, I refer to the basic joint-venture model presented in the literature [27] (the JV-B model). In the JV-B model, two agents $i$ and $j$ ($i{\neq}j,i,j = 1,2,\cdots,N$) are randomly selected at time $t$ among $N$ agents. Both agents have wealth $m_i(t)$ and $m_j(t)$, respectively, and a common savings rate, $\lambda$. Both agents contribute their wealth, excluding savings, to the joint venture, and wealth is distributed according to the wealth $(1-\lambda){\cdot}m_i(t)$ and $(1-\lambda){\cdot}m_j(t)$ contributed by each agent and the profit/loss ratio $\delta$. The wealth $m_i(t+1)$ and $m_j(t+1)$ of agents $i$ and $j$ at time $t+1$ are expressed as Eq (2), respectively.

$$m_i(t + 1) = \lambda \cdot m_i(t) + (1 + \delta) \cdot (1 - \lambda) \cdot m_i(t);$$

$$m_j(t + 1) = \lambda \cdot m_j(t) + (1 + \delta) \cdot (1 - \lambda) \cdot m_j(t). \tag{2}$$

In the JV-B model, all wealth, excluding savings, contributes to the joint venture. But in the joint-venture model in this study, the two agents $i$ and $j$ contributing wealth according to their moral responsibilities $\rho_{Mi}$ and $\rho_{Mj}$ are modeled (the JV-M model). That is, two agents $i$ and $j$ contribute wealth $(1-\lambda)\cdot\rho_{Mi}\cdot m_i(t)$ and $(1-\lambda)\cdot\rho_{Mj}\cdot m_j(t)$, respectively, and wealth is distributed according to the profit/loss rate $\delta$. Wealth $m_i(t+1)$ and $m_j(t+1)$ at time $t+1$ are expressed in Eq (3).

$$m_i(t+1) = \lambda \cdot m_i(t) + (1-\lambda) \cdot (1 - \rho_{Mi}) \cdot m_i(t)$$

$$+(1+\delta) \cdot (1-\lambda) \cdot \rho_{Mi} \cdot m_i(t);$$

$$m_j(t+1) = \lambda \cdot m_j(t) + (1-\lambda) \cdot (1 - \rho_{Mj}) \cdot m_j(t)$$

$$+(1+\delta) \cdot (1-\lambda) \cdot \rho_{Mj} \cdot m_j(t). \tag{3}$$

Next, a redistribution model for the JV-M model (the JV-M-M model) is formulated by referring to the redistribution model in the literature [27]. In the JV-M-M model, each of the $N$ agents contributes wealth $\xi\cdot\rho_{Mi}\cdot m_i(t)$ according to its transfer ratio, $\xi$, and moral responsibility, $\rho_{Mi}$, every redistribution period, $t_p$, and the wealth $\sum_{k=1}^{N} \rho_{Mk} \cdot m_k(t)$ collected from the $N$ agents is redistributed to each agent according to its moral responsibility ratio, $\rho_{Mi}/\Sigma_k\rho_{Mk}$. Here $k$ is the index of summation. The usage of $k$ is the same in the following cases. The wealth $m_i(t+\Delta)$ of agent $i$ at time $t+\Delta$ after redistribution is expressed in Eq (4).

$$m_i(t+\Delta) = (1 - \xi \cdot \rho_{Mi}) \cdot m_i(t) + \xi \cdot \frac{\rho_{Mi}}{\sum_k \rho_{Mk}} \cdot \sum_{k=1}^{N} \rho_{Mk} \cdot m_k(t). \tag{4}$$

In the JV-M-M model, redistribution was made according to moral responsibility; for comparison, I model redistribution according to agent $i$'s risk vulnerability, $\rho_{Ri}$, (called the JV-M-R model). In the JV-M-R model, the wealth collected from $N$ agents is redistributed to each of them according to their risk vulnerability ratio, $\rho_{Ri}/\Sigma_k\rho_{Rk}$. The wealth $m_i(t+\Delta)$ of agent $i$ at time $t+\Delta$ after redistribution is expressed in Eq (5).

$$m_i(t+\Delta) = (1 - \xi \cdot \rho_{Mi}) \cdot m_i(t) + \xi \cdot \frac{\rho_{Ri}}{\sum_k \rho_{Rk}} \cdot \sum_{k=1}^{N} \rho_{Mk} \cdot m_k(t). \tag{5}$$

Similar to the JV-M-M and JV-M-R models, I model the redistribution according to both moral responsibility, $\rho_{Mi}$, and risk vulnerability, $\rho_{Ri}$, of agent $i$ (called the JV-M-MR model). In the JV-M-MR model, the wealth collected from $N$ agents is redistributed to each according to the ratio $\rho_i/\Sigma_k\rho_k$ using $\rho_i = \rho_{Mi}\cdot\rho_{Ri}$ in Eq (1). The wealth $m_i(t+\Delta)$ of agent $i$ at time $t+\Delta$ after redistribution is expressed in Eq (6).

$$m_i(t+\Delta) = (1 - \xi \cdot \rho_{Mi}) \cdot m_i(t) + \xi \cdot \frac{\rho_i}{\sum_k \rho_k} \cdot \sum_{k=1}^{N} \rho_{Mk} \cdot m_k(t). \tag{6}$$

## WE economy models

In the JV-M model, wealth is distributed according to the wealth $(1-\lambda)\cdot\rho_{Mi}\cdot m_i(t)$ and $(1-\lambda)\cdot\rho_{Mj}\cdot m_j(t)$ contributed by agents $i$ and $j$, respectively. In the WE economy, to distribute as co-adventurers or a community of destiny, the wealth contributed by agents $i$ and $j$ according to their moral responsibilities $\rho_{Mi}$ and $\rho_{Mj}$ is collected once as $(1-\lambda) \cdot (\rho_{Mi} \cdot m_i(t) + \rho_{Mj} \cdot m_j(t))$ and distributed according to their respective moral responsibility ratios $\rho_{Mi}/(\rho_{Mi}+\rho_{Mj})$ and

$\rho_{Mj}/(\rho_{Mi}+\rho_{Mj})$ (called the WE-M-M model). The wealth $m_i(t+1)$ and $m_j(t+1)$ of agents $i$ and $j$ at time $t+1$ are expressed in Eq (7), respectively.

$$m_i(t+1) = \lambda \cdot m_i(t) + (1-\lambda) \cdot (1-\rho_{Mi}) \cdot m_i(t)$$

$$+(1+\delta) \cdot (1-\lambda) \cdot \frac{\rho_{Mi}}{\rho_{Mi} + \rho_{Mj}} \cdot (\rho_{Mi} \cdot m_i(t) + \rho_{Mj} \cdot m_j(t));$$

$$m_j(t+1) = \lambda \cdot m_j(t) + (1-\lambda) \cdot (1-\rho_{Mj}) \cdot m_j(t)$$

$$+(1+\delta) \cdot (1-\lambda) \cdot \frac{\rho_{Mj}}{\rho_{Mi} + \rho_{Mj}} \cdot (\rho_{Mi} \cdot m_i(t) + \rho_{Mj} \cdot m_j(t)). \tag{7}$$

In the WE-M-M model, redistribution was made according to moral responsibility; for comparison, I model redistribution according to agent $i$'s risk vulnerability $\rho_{Mi}$ (called the WE-M-R model). In the WE-M-R model, the wealth contributed by the two agents $i$ and $j$ according to their moral responsibilities, $\rho_{Mi}$ and $\rho_{Mj}$, is distributed according to their risk vulnerability ratios, $\rho_{Ri}/(\rho_{Ri}+\rho_{Rj})$ and $\rho_{Rj}/(\rho_{Ri}+\rho_{Rj})$, respectively. Wealth $m_i(t+1)$ and $m_j(t+1)$ at time $t+1$ are expressed in Eq (8).

$$m_i(t+1) = \lambda \cdot m_i(t) + (1-\lambda) \cdot (1-\rho_{Mi}) \cdot m_i(t)$$

$$+(1+\delta) \cdot (1-\lambda) \cdot \frac{\rho_{Ri}}{\rho_{Ri} + \rho_{Rj}} \cdot (\rho_{Mi} \cdot m_i(t) + \rho_{Mj} \cdot m_j(t));$$

$$m_j(t+1) = \lambda \cdot m_j(t) + (1-\lambda) \cdot (1-\rho_{Mj}) \cdot m_j(t)$$

$$+(1+\delta) \cdot (1-\lambda) \cdot \frac{\rho_{Rj}}{\rho_{Ri} + \rho_{Rj}} \cdot (\rho_{Mi} \cdot m_i(t) + \rho_{Mj} \cdot m_j(t)). \tag{8}$$

Similar to the WE-M-M and WE-M-R models, I model distribution according to both the moral responsibilities $\rho_{Mi}$, $\rho_{Mj}$ and risk vulnerabilities $\rho_{Ri}$, $\rho_{Rj}$ of the two agents $i$ and $j$ (called the WE-M-MR model). In the WE-M-MR model, using $\rho_i = \rho_{Mi} \cdot \rho_{Ri}$ in Eq (1), the wealth contributed by agents $i$ and $j$ according to their moral responsibilities $\rho_{Mi}$ and $\rho_{Mj}$ is distributed according to their ratios $\rho_i/(\rho_i+\rho_j)$ and $\rho_j/(\rho_i+\rho_j)$, respectively. Wealth $m_i(t+1)$ and $m_j(t+1)$ at time $t+1$ are expressed in Eq (9).

$$m_i(t+1) = \lambda \cdot m_i(t) + (1-\lambda) \cdot (1-\rho_{Mi}) \cdot m_i(t)$$

$$+(1+\delta) \cdot (1-\lambda) \cdot \frac{\rho_i}{\rho_i + \rho_j} \cdot (\rho_{Mi} \cdot m_i(t) + \rho_{Mj} \cdot m_j(t));$$

$$m_j(t+1) = \lambda \cdot m_j(t) + (1-\lambda) \cdot (1-\rho_{Mj}) \cdot m_j(t)$$

$$+(1+\delta) \cdot (1-\lambda) \cdot \frac{\rho_j}{\rho_i + \rho_j} \cdot (\rho_{Mi} \cdot m_i(t) + \rho_{Mj} \cdot m_j(t)). \tag{9}$$

## Impact of free riders

To examine the impact of free riders who are not cooperative in joint ventures and WE economies, I refer to the JV-M model in Eq (3) and WE-M-M model in Eq (7). Assuming that one agent, $j$, of the two agents contributes wealth only by multiplying its moral responsibility, $\rho_{Mj}$, by the ratio, $r_f$, Eqs (3) and (7) can be rewritten as Eqs (10) and (11), respectively. For convenience, I call to the model combining Eq (10) with the redistribution in Eq (4) the JV-M-M-FR model and the model in Eq (11) the WE-M-M-FR model. Note that redistribution in the JV-M-M-FR model does not consider the impact of free riders because redistribution is institutionally done for everyone.

$$m_i(t+1) = \lambda \cdot m_i(t) + (1-\lambda) \cdot (1-\rho_{Mi}) \cdot m_i(t)$$

$$+(1+\delta) \cdot (1-\lambda) \cdot \rho_{Mi} \cdot m_i(t);$$

$$m_j(t+1) = \lambda \cdot m_j(t) + (1-\lambda) \cdot (1-r_f \cdot \rho_{Mj}) \cdot m_j(t)$$

$$+(1+\delta) \cdot (1-\lambda) \cdot r_f \cdot \rho_{Mj} \cdot m_j(t). \tag{10}$$

$$m_i(t+1) = \lambda \cdot m_i(t) + (1-\lambda) \cdot (1-\rho_{Mi}) \cdot m_i(t)$$

$$+(1+\delta) \cdot (1-\lambda) \cdot \frac{\rho_{Mi}}{\rho_{Mi}+\rho_{Mj}} \cdot (\rho_{Mi} \cdot m_i(t) + r_f \cdot \rho_{Mj} \cdot m_j(t));$$

$$m_j(t+1) = \lambda \cdot m_j(t) + (1-\lambda) \cdot (1-r_f \cdot \rho_{Mj}) \cdot m_j(t)$$

$$+(1+\delta) \cdot (1-\lambda) \cdot \frac{\rho_{Mj}}{\rho_{Mi}+\rho_{Mj}} \cdot (\rho_{Mi} \cdot m_i(t) + r_f \cdot \rho_{Mj} \cdot m_j(t)). \tag{11}$$

## Calculation of Gini index

The Gini index is a well-known parameter for assessing wealth inequality [30] and is calculated by drawing a Lorenz curve and equal distribution line [31]. There are various inequality indices calculated from Lorenz curves [32], however, this study uses the Gini index. By the operation $Sort(m_i(t))$ in Eq (12), the wealth $m_i(t)$ ($i = 1,2,\cdots,N$) of the $N$ agents at time $t$ is sorted from smallest to largest, and the $k$-th wealth, from the smallest, is set as $\mu_k(t)$, and the Gini index $g$ is calculated.

$$\mu_k(t) \in Sort(m_i(t)),$$

$$g = \frac{2 \cdot \sum_{k=1}^{N} k \cdot \mu_k(t)}{N \cdot \sum_{k=1}^{N} \mu_k(t)} - \frac{N+1}{N}. \tag{12}$$

The Gini index equals zero if the wealth of $N$ agents is equally distributed, and one if the wealth is delta-distributed (all wealth is concentrated in only one agent). In other words, the more unequal the distribution, the larger the Gini index.

## Results

First, the common parameters for the simulations of the joint-venture, redistribution, and WE economy models were set. The number of agents is $N = 1,000$ (which does not affect the

relative calculation of wealth distribution or Gini index), and time is run from $t = 0$ to $10^6$ in increments of 1. The initial distribution of the wealth of the $N$ agents at time $t = 0$ is equal to $m_i(0) = 1$ ($i = 1,2,\cdots,N$). The savings rate is $\lambda = 0.25$, referring to the world's gross savings (as a percentage of GDP) [33]. With respect to the profit/loss ratio $\delta$, while the average return of the stock index is about 8%, there are large fluctuations exceeding ±10% [34], compared to an average return of only about 2% for investors [35]. Therefore, to account for the fact that business profits and losses fluctuate both positively and negatively, this study sets a uniform random number in the range $-0.1 \leq \delta \leq 0.1$ ($\delta_w = 0.1$) for every time $t$ for the profit/loss rate $\delta$. For the redistribution period, $t_p$, and transfer rate, $\xi$, of the joint-venture model, I use the combination $t_p = 10^4$ and $\xi = 0.5$, where the Gini index $g$ is relatively small, referring to the literature [27].

Fig 1A and 1B show the calculation results for the JV-B model of conventional joint ventures expressed in Eq (1); Fig 1C and 1D show the JV-M model of joint ventures based on moral responsibility expressed in Eq (2); and Fig 1E and 1F show the WE-M-M model of the WE economy based on moral responsibility expressed in Eq (7). Fig 1A, 1C and 1E show the frequency distribution at times $t = 10^4$, $10^5$, and $10^6$, respectively, with wealth $m$ on the horizontal axis and frequencies on the vertical axis. Fig 1B, 1D and 1F plot the wealth $m$ of each agent at time $t = 10^6$ with agent number, #, on the horizontal axis.

In the JV-B and JV-M models, as time $t$ increases, the frequency distributions of wealth in Fig 1A and 1C approach the $m = 0$ side, whereas the wealth $m$ of the agents in Fig 1B and 1D are widely distributed. The literature [27] shows that the joint-venture model without redistribution gradually approaches a delta distribution (Gini index $g = 1$). In Fig 1D, there are no plots near $m = 0$ for # roughly in the range of 1 to 300. This is because, in the JV-M model, the smaller the # is, the smaller the moral responsibility; thus, the wealth contribution is suppressed and less subject to fluctuations in profits and losses. Compared to the JV-B and JV-M models, the WE-M-M model in Fig 1E and 1F concentrates on the distribution of wealth near $m = 1$. This is because, as can be seen by comparing Eqs (3) and (7), in the WE-M-M model, the wealth contributed by the two agents is added together and then distributed according to the ratio of moral responsibility.

Fig 2 shows the calculation results for the combinations of joint-venture models and redistribution models. Fig 2A and 2B show the JV-M-M model combining Eqs (3) and (4), Fig 2C and 2D show the JV-M-R model combining Eqs (3) and (5), and Fig 2E and 2F show the JV-M-MR model combining Eqs (3) and (6). Compared with the JV-M models in Fig 1C and 1D, in Fig 2A, 2B, 2E and 2F, the distribution range of wealth $m$ is narrower owing to redistribution and concentrated near $m = 1$. In Fig 2B and 2F, the variance in wealth $m$ is larger for larger # because larger wealth contributions are subject to fluctuations in profits and losses. In contrast to Fig 2B, in Fig 2F, wealth $m$ on the side with the smaller # is slightly larger than that on the side with a larger # because of redistribution based on both moral responsibility and risk vulnerability. In Fig 2C, the frequency distribution is skewed toward $m = 0$, and in Fig 2D, the wealth $m$ is smaller for larger #. This is because, as Eqs (1) and (5) show, wealth contributed according to moral responsibility was redistributed according to risk vulnerability, resulting in an imbalance between the contribution and redistribution of wealth.

Fig 3 shows the calculation results for the WE economy models. Fig 3A and 3B show the WE-M-M model of Eq (7) (the same as Fig 1E and 1F, respectively), Fig 3C and 3D show the WE-M-R model of Eq (8), and Fig 3E and 3F show the WE-M-MR model of Eq (9). Compared with the joint-venture and redistribution models in Fig 2A, 2B, 2E and 2F, the WE economy models in Fig 3A, 3B, 3E and 3F further concentrate on the distribution of wealth $m$ near $m = 1$. In other words, WE economies have the advantage of reducing inequality in respect to joint ventures. This is due to the wealth addition effect in WE economies, as shown in Fig 1.

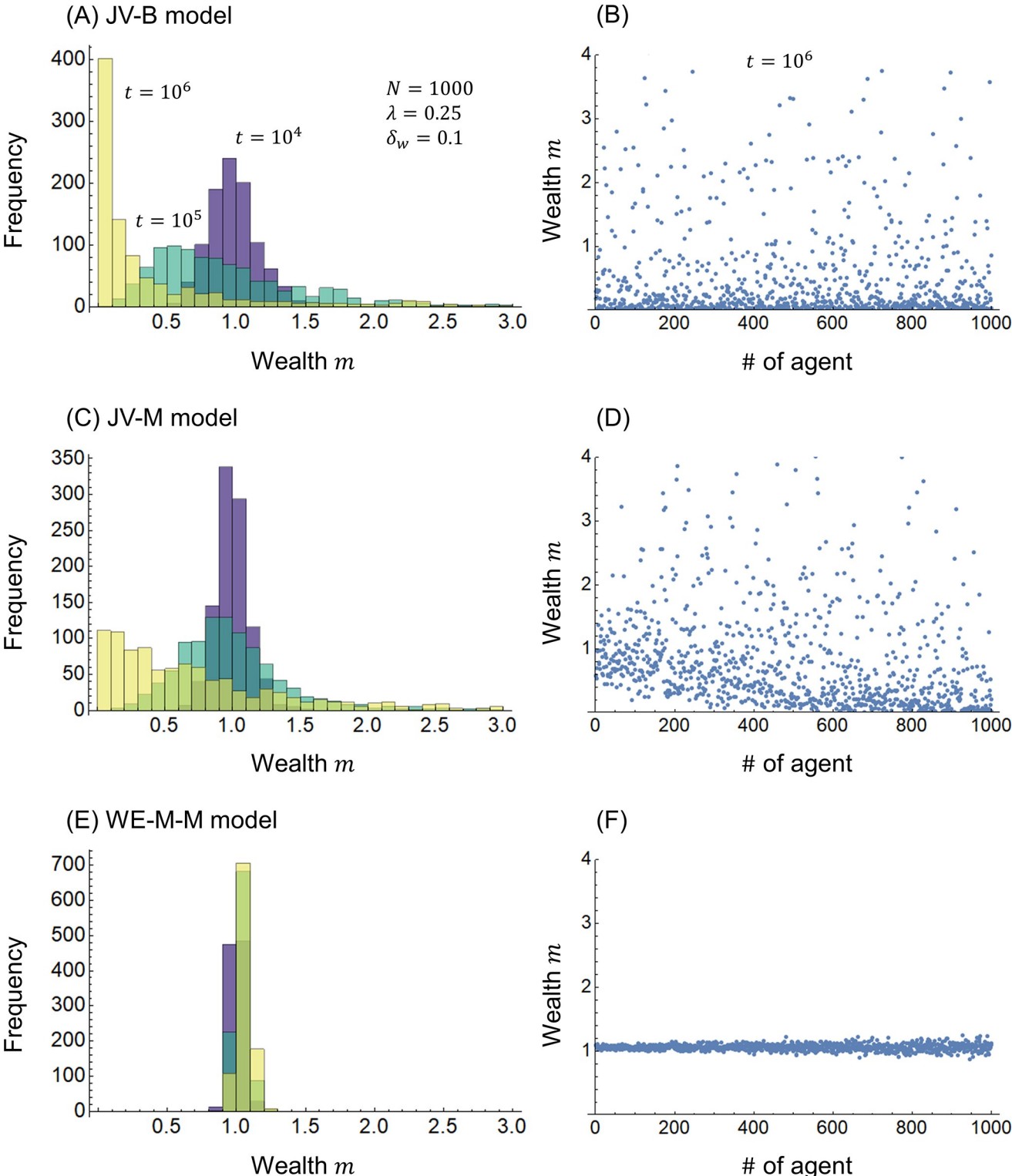

**Fig 1. Wealth distribution.** (A)(B) JV-B model, (C)(D) JV-M model, and (E)(F) WE-M-M model.

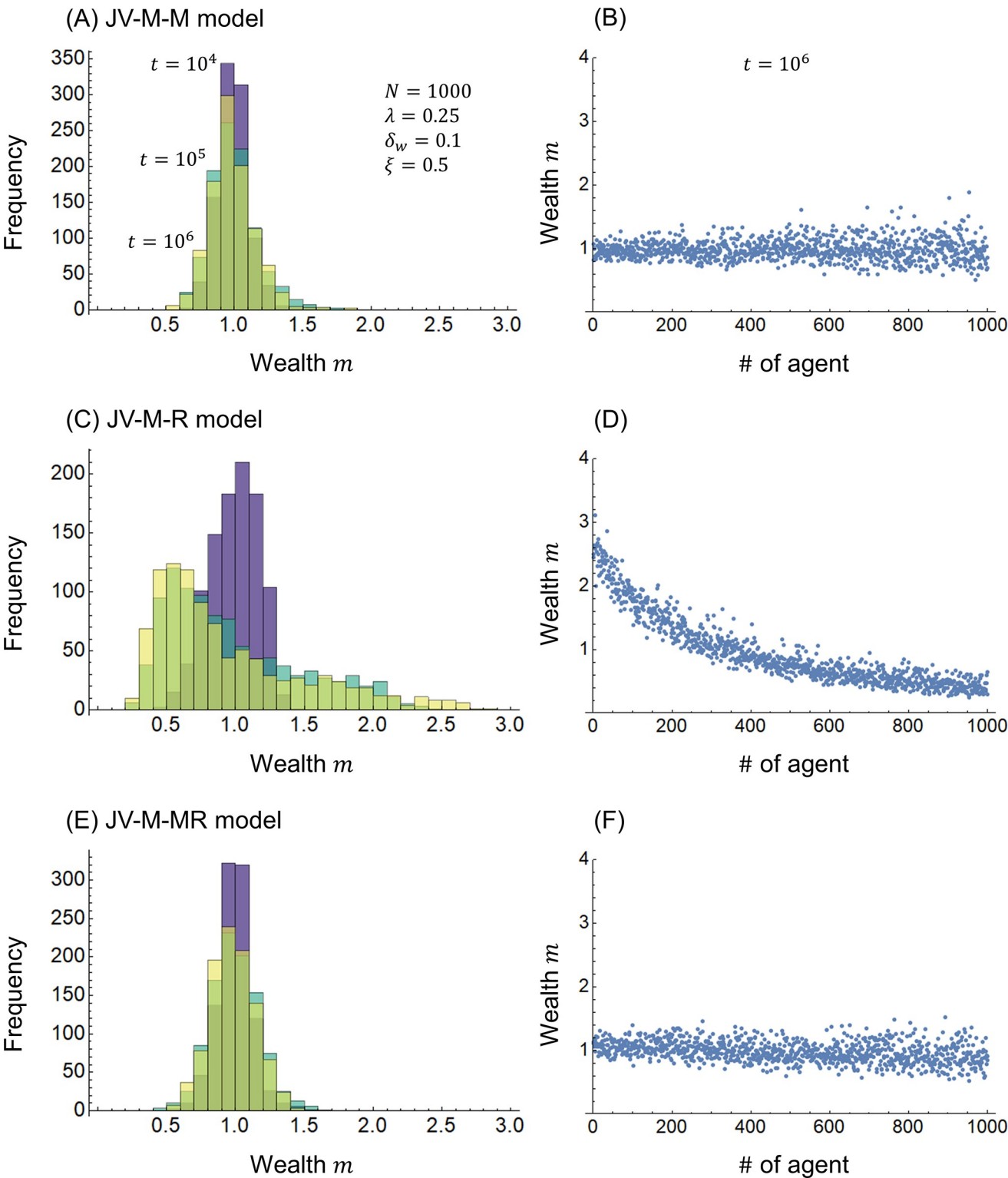

**Fig 2. Wealth distribution.** (A)(B) JV-M-M model, (C)(D) JV-M-R model, and (E)(F) JV-M-MR model.

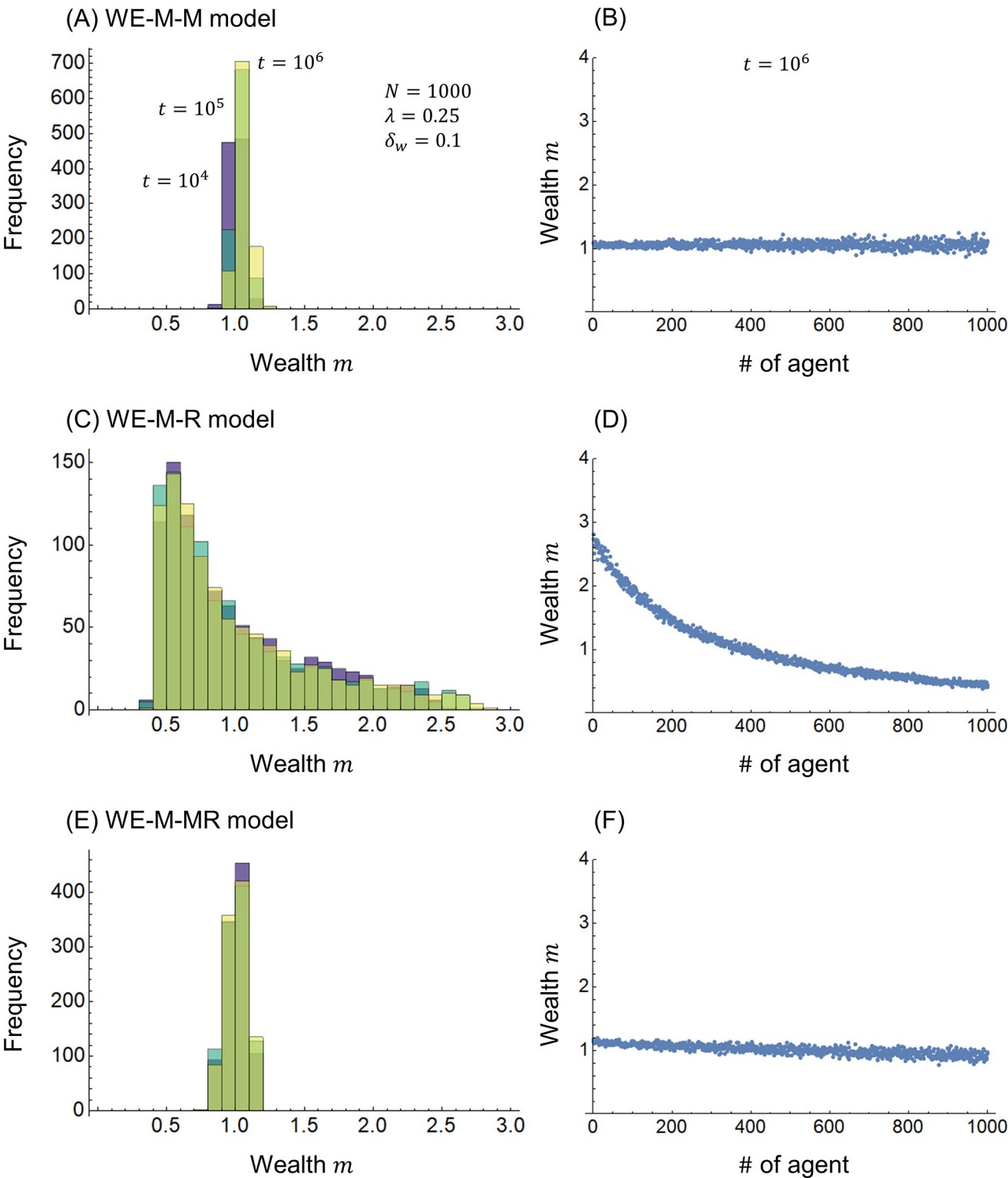

**Fig 3. Wealth distribution.** (A)(B) WE-M-M model, (C)(D) WE-M-R, and (E)(F) WE-M-MR model.

The skewed distribution of wealth $m$ in Fig 3C and 3D is due to the imbalance between the contribution and distribution of wealth, as shown in Fig 2C and 2D. The difference between Fig 3B and 3F is due to the inclusion of risk vulnerability in the distribution, which is similar to the difference between Fig 2B and 2F. Note that Figs 2B and 3B, Figs 2D and 3D, and Figs 2E and 3E show nearly equivalent trends, suggesting that the joint-venture redistribution and WE economy distribution have similar inequality reduction effects.

Fig 4 shows the calculation results when the initial distribution of wealth $m_i(0)$ ($i = 1,2,\cdots$, $N$) is changed from an even distribution with all 1s to a real uniform random number distribution between 0 and 2. Fig 4A and 4B show the JV-M-M-IR model with a modified initial distribution of the JV-M-M model, and Fig 4C and 4D show the WE-M-M-IR model with a modified initial distribution of the WE-M-M model. The JV-M-M-IR model in Fig 4A and 4B for the JV-M-M model in Fig 2A and 2B, and the WE-M-M-IR model in Fig 4C and 4D for the WE-M-M model in Fig 3A and 3B show almost equivalent results. This result indicates that the redistribution of joint ventures and WE economies has the resilience to converge the distribution of wealth $m$ over time $t$. Moreover, the resilience of WE economies is higher than that of the redistribution of joint ventures.

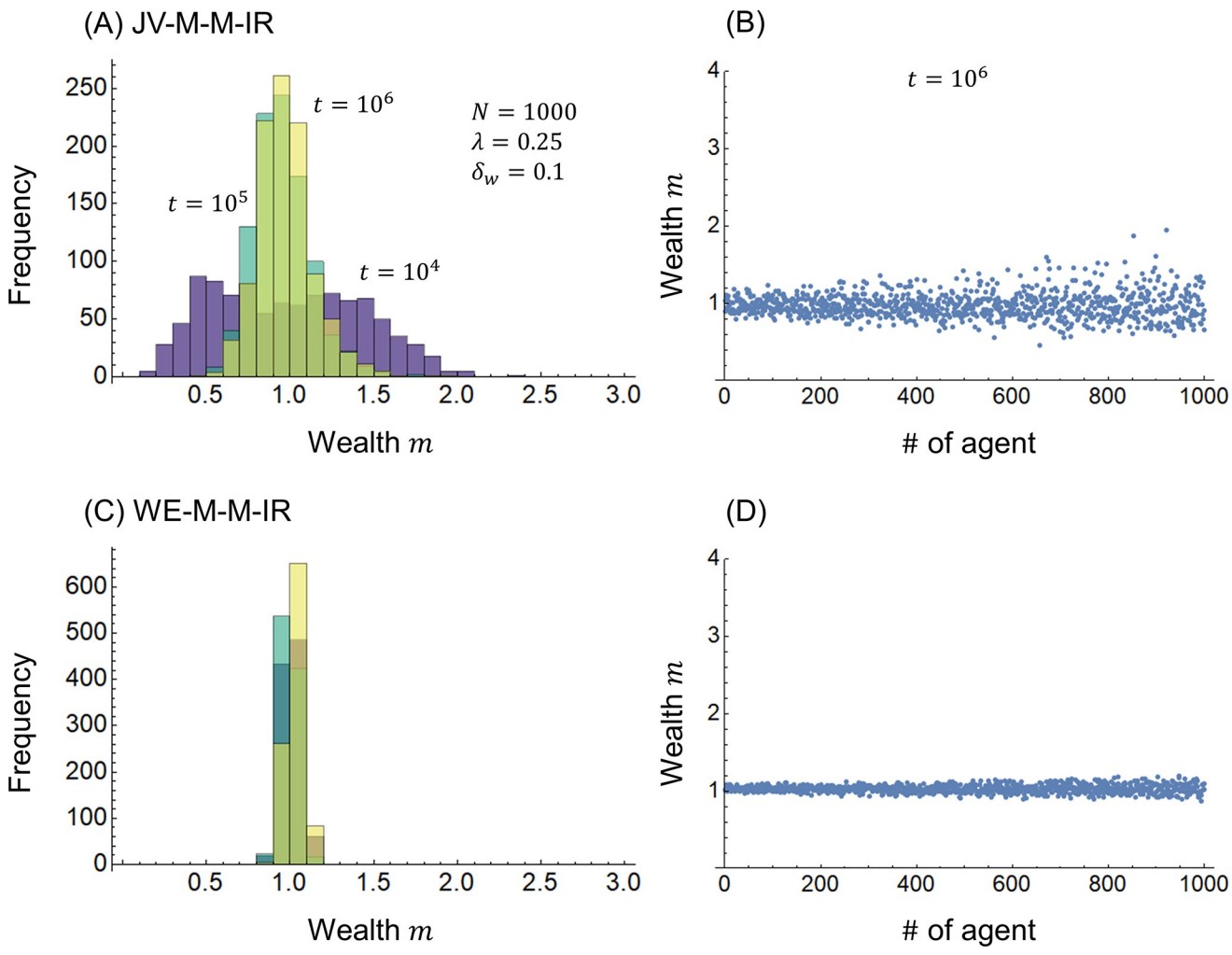

**Fig 4. Wealth distribution.** (A)(B) JV-M-M-IR model and (C)(D) WE-M-M-IR model.

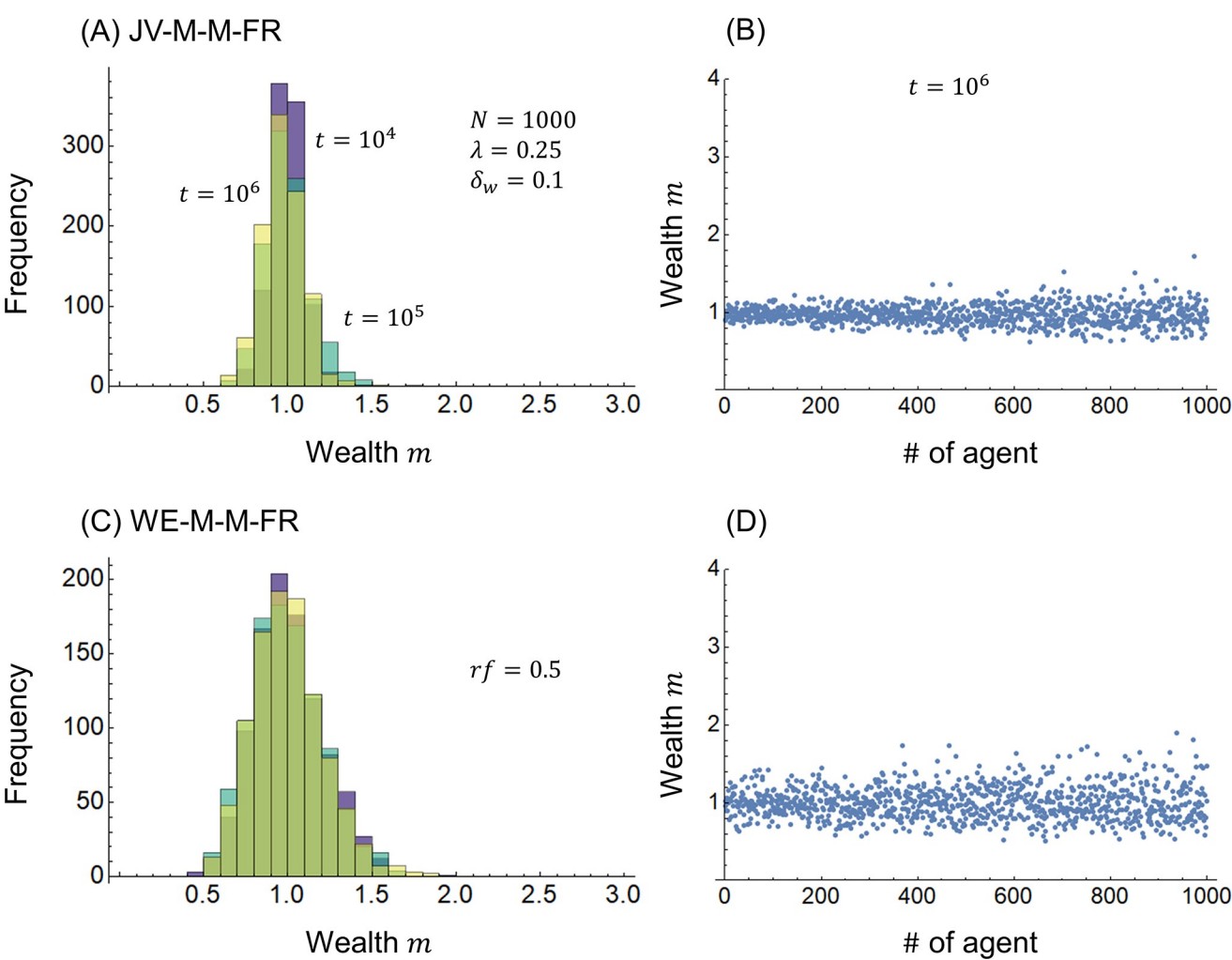

**Fig 5. Wealth distribution.** (A)(B) JV-M-M-FR model and (C)(D) WE-M-M-FR model.

Fig 5 shows the calculation results when free riders are considered. The ratio of free riders in Eqs (10) and (11) is assumed to be $r_f = 0.5$ simply for calculation. Fig 5A and 5B show the JV-M-M-FR model combining Eq (10) and the redistribution in Eq (4), whereas Fig 5C and 5D show the WE-M-M-FR model in Eq (11). Comparing the JV-M-M model in Fig 2A and 2B with the JV-M-M-FR model in Fig 5A and 5B, there is almost no difference in the distribution of wealth $m$ between the two models. This is because, as shown in Eq (10), in the joint venture model, the effect of a free-rider agent only spills over to the agent itself. By contrast, the WE-M-M-FR model in Fig 5C and 5D has a wider distribution of wealth $m$ than the WE-M-M model in Fig 3A and 3B. As shown in Eq (11), in the WE economy model, a reduction in the free-rider agent's wealth contribution affects both agents. This indicates that WE economies have the disadvantage of being more susceptible to free riders than joint ventures.

Fig 6 shows the results of the Gini index calculations. Fig 6 shows the change in the Gini index $g$ with time $t$, where the horizontal axis is time $t$ and the vertical axis (logarithm) is the Gini index $g$. In Fig 6A, the Gini index $g$ of the JV-M model without redistribution (the gray line) tends toward 1 with time $t$. The Gini indices $g$ for the joint-venture model with redistribution and WE economy model converge to a constant value less than the social unrest

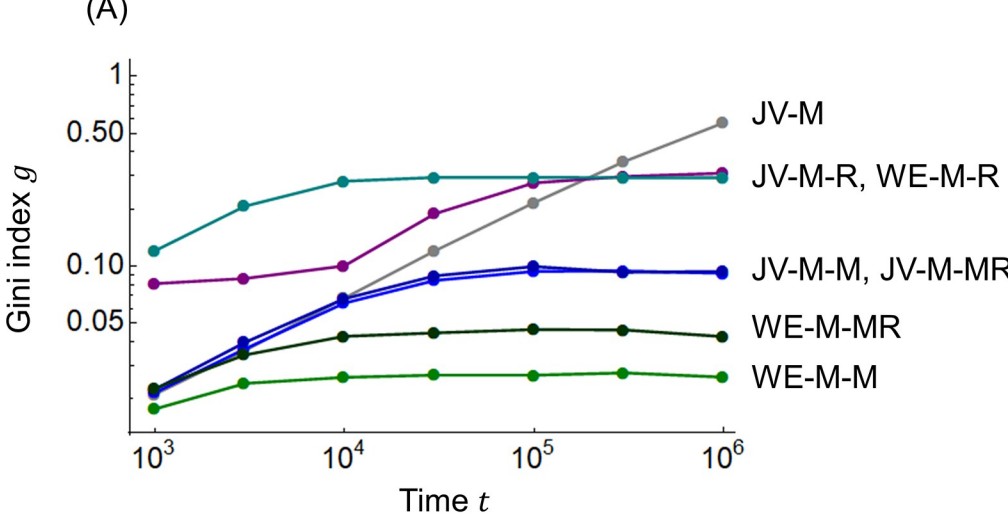

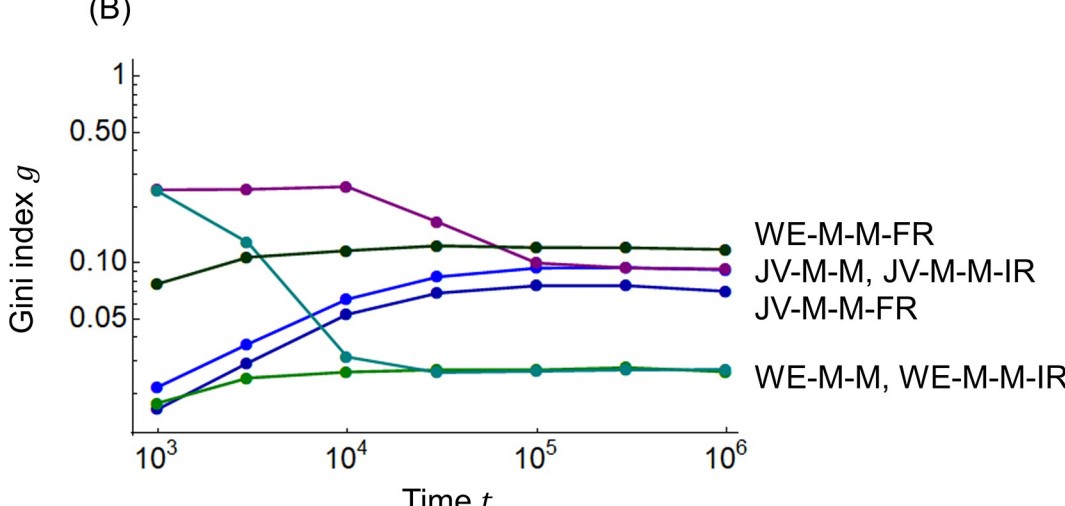

**Fig 6. Gini index.** (A) JV-M, JV-M-M, JV-M-R, JV-M-MR; WE-M-M, WE-M-R, and WE-M-MR models. (B) JV-M-M, JV-M-M-IR, JV-M-M-FR, WE-M-M, WE-M-M-IR, and WE-M-M-FR models.

warning level of 0.4 [3]. The WE-M-M model (green) is the smallest, followed by the WE-M-MR model (dark green), which includes risk vulnerability. The JV-M-M (blue) and JV-M-MR (dark blue) models, and the JV-M-R (purple) and WE-M-R (blue-green) models have nearly equivalent values, respectively. The overall trend is that the WE economy model has a smaller Gini index than the joint venture model, and the redistribution of joint ventures and WE economies, including risk vulnerability with moral responsibility, has a slightly larger Gini index. However, the large Gini index in the model that encapsulates moral responsibility and risk vulnerability is a positive inequality to consider future generations and the socially vulnerable.

In Fig 6B, the JV-M-M (blue) and JV-M-M-IR (purple) models, and the WE-M-M (green) and WE-M-M-IR (blue-green) models, which have different initial distributions of wealth *m*, converge to approximately the same value of the Gini index *g* over time *t*, respectively. Concerning the impact of free riders, the Gini index *g* is slightly smaller in the JV-M-M-FR model

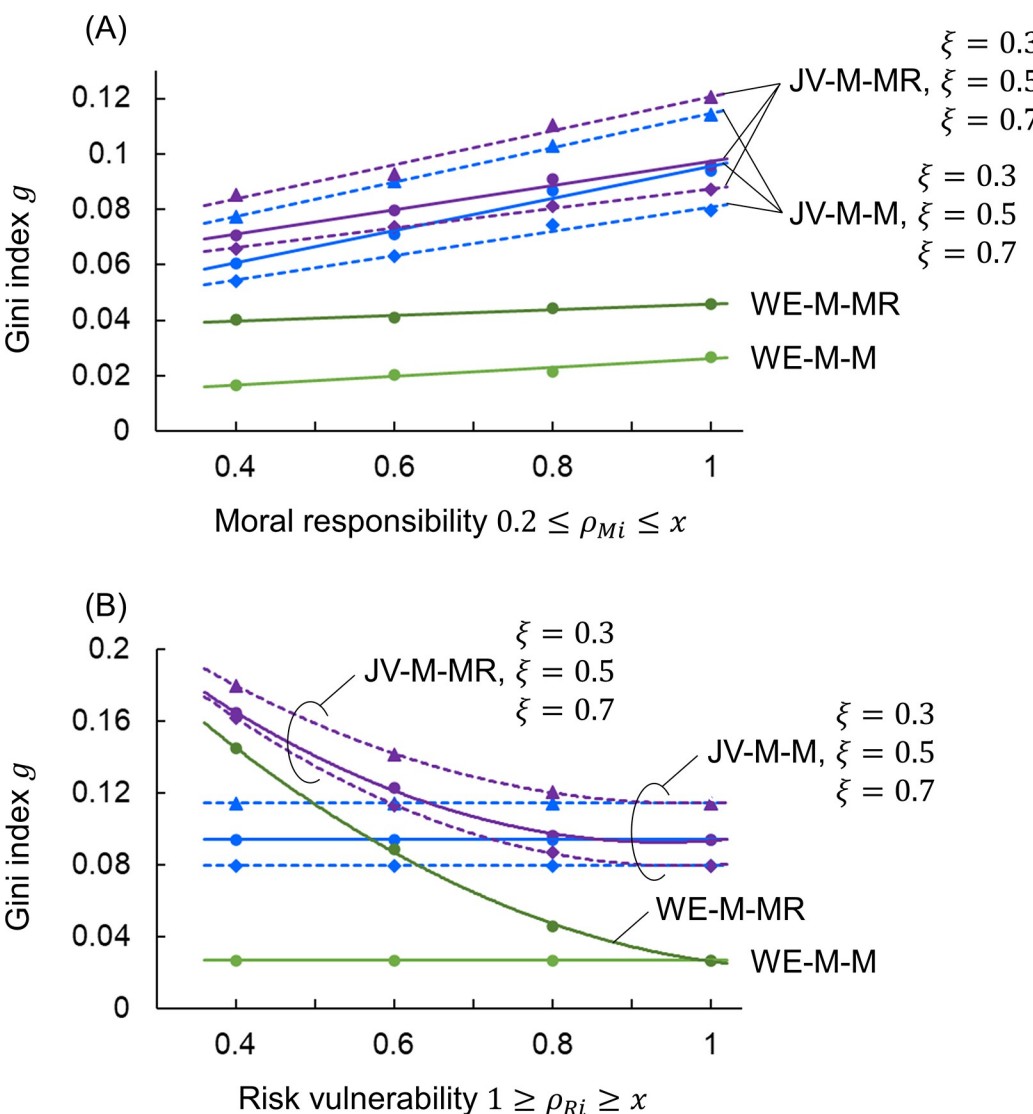

**Fig 7. Gini index sensitivity for parameter variations of JV-M-M, JV-M-MR, WE-M-M, WE-M-MR models.** (A) the range of $\rho_{Mi}$ and $\xi$. (B) the range of $\rho_{Ri}$ and $\xi$.

(dark blue) than in the JV-M-M model (blue). This is because of the reduced contribution of free-rider wealth, which reduces the impact of profits and losses. This has been shown in the literature [26, 28] as a proportional relationship between the amount of contributions and the Gini index (i.e., the Gini index decreases as the amount of contributions is reduced). The Gini index $g$ of the WE-M-M-FR model (dark green) is larger than that of the WE-M-M model (green). This has previously been explained as the reason for the difference between Fig 5C and 5D relative to Fig 3A and 3B.

Figs 7 and 8 show results for the joint venture/redistribution and WE economy models, examining the sensitivity of the Gini index to variations in the parameters of those models. For the basic setup of moral responsibility $0.2{\leq}\rho_{Mi}{\leq}1$, risk vulnerability $1{\geq}\rho_{Ri}{\geq}0.8$, redistribution period $t_p = 10^4$, transfer rate $\xi = 0.5$, savings rate $\lambda = 0.25$, and profit/loss rate range $\delta_w = 0.1$, Fig 7A changes the range of $\rho_{Mi}$ and $\xi$, Fig 7B changes the range of $\rho_{Ri}$ and $\xi$, Fig 8A changes $\lambda$, and Fig 8B changes $\delta_w$.

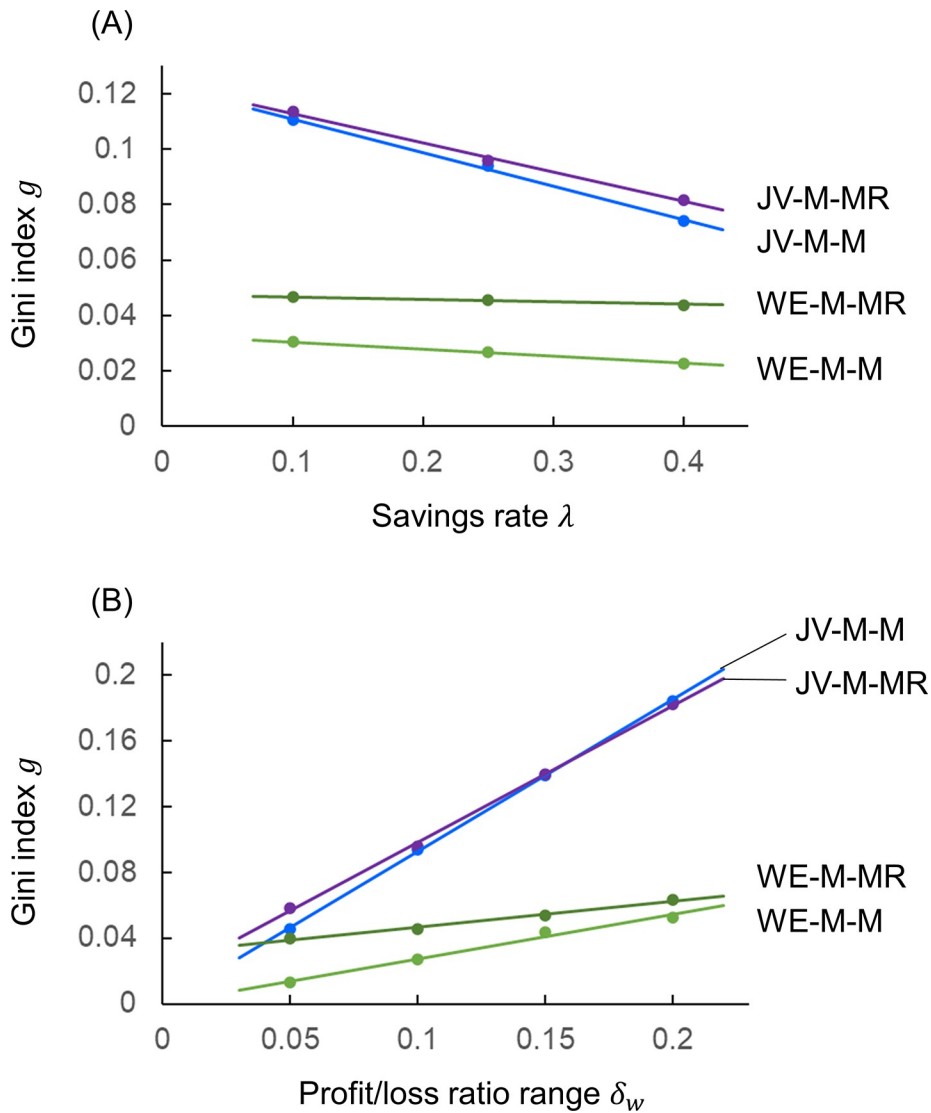

**Fig 8. Gini index sensitivity for parameter variations of JV-M-M, JV-M-MR, WE-M-M, WE-M-MR models.** (A) $\lambda$. (B) $\delta_w$.

In Fig 7A, the horizontal axis is the range of $\rho_{Mi}$ and the vertical axis is the Gini index $g$. By changing the constant 0.8 in the formula for $\rho_{Mi}$ in Eq (1) from 0.2 to 0.8, the upper limit of the range of $\rho_{Mi}$ is changed from 0.4 to 1. The transfer rates $\xi$ are set to 0.3, 0.5, and 0.7, which are equivalent to setting $t_p$ to $1.67\times10^4$, $10^4$, and $0.71\times10^4$ respectively, because the effect of the redistribution of joint ventures depends on $\xi/t_p$ [28]. The JV-M-M (blue), JV-M-MR (purple), WE-M-M (green), and WE-M-MR models (dark green) have smaller Gini indices $g$ as the upper limit of $\rho_{Mi}$ decreases. This is because the narrower the range of $\rho_{Mi}$, the smaller the difference between agents. The difference in the Gini indices between the JV-M-M and JV-M-MR models and the WE-M-M and WE-M-MR models is due to the differences in wealth distribution between joint ventures (Eq (3)) and WE economies (Eqs (7) and (9)), as described in Fig 1. In the JV-M-M and JV-M-MR models, the Gini index $g$ is smaller as the transfer rate $\xi$ increases, as expected.

In Fig 7B, the horizontal axis is the range of $\rho_{Ri}$ and the vertical axis is the Gini index $g$. By changing the constant 0.2 in the formula for $\rho_{Ri}$ in Eq (1) from 0.6 to 0, the lower limit of the range of $\rho_{Ri}$ is changed from 0.4 to 1. The transfer rates $\xi$ are set to 0.3, 0.5, and 0.7, as in Fig 7A. In the JV-M-M (blue) and the WE-M-M (green) models, Eqs (3), (4), and (7) are independent of risk vulnerability $\rho_{Ri}$; therefore, the Gini index $g$ is unchanged. In both the JV-M-MR model (purple) and the WE-M-MR model (dark green), the Gini index $g$ increases as the lower limit of $\rho_{Ri}$ decreases. This is because, as described in Fig 6A, the range of $\rho_{Ri}$ is widened to weight redistribution and distribution to the vulnerable, giving positive inequality. Both graphs are curvilinear because $\rho_{Ri}$ is involved in the fractional terms in Eqs (6) and (9). In the JV-M-M and JV-M-MR models, as in Fig 7A, the Gini index $g$ decreases as the transfer rate $\xi$ increases.

In Fig 8A the horizontal axis is the savings rate $\lambda$, in Fig 8B the horizontal axis is the profit/loss rate range $\delta_w$, and the vertical axis for both is the Gini index $g$. In Fig 8A, as shown in Eq (3) for the JV-M-M (blue) and JV-M-MR (purple) models and Eqs (7) and (9) for the WE-M-M (green) and WE-M-MR (dark green) models, as the savings rate $\lambda$ increases, the wealth exchange decreases, and thus the Gini index $g$ becomes smaller. Incidentally, $g = 0$ for $\lambda = 1$. In Fig 8B, the Gini index $g$ increases as the profit/loss rate range $\delta_w$ increases for the JV-M-M, JV-M-MR, WE-M-M, and WE-M-MR models. This is because, as inferred in Figs 2B, 2F, 3B and 3F, the variation in profit/loss is larger for agents with larger moral responsibility $\rho_{Mi}$. As $\delta_w$ increases, the difference in $g$ between the JV-M-M and JV-M-MR models and between the WE-M-M and WE-M-MR models decreases. This is because a large $\delta_w$ is speculative and undesirable, but in the JV-M-MR and WE-M-MR models, the wealth of agents with a large moral responsibility $\rho_{Mi}$ is redistributed or distributed to the vulnerable, thereby reducing inequality. Figs 7 and 8 show that the joint venture/redistribution model and the WE economy model retain the effect of reducing inequality for variations in the model parameters, i.e., both have robustness to diverse economic environments.

Fig 9 transcribes the historical findings on the Gini index in the Old World (from approximately 11,000 to 2,000 years ago) and the New World (from approximately 3,000 to 300 years ago) in the post-Neolithic period [36] to show the position of the WE economy and joint venture/redistribution models. The literature measures the Gini index using house size as a proxy

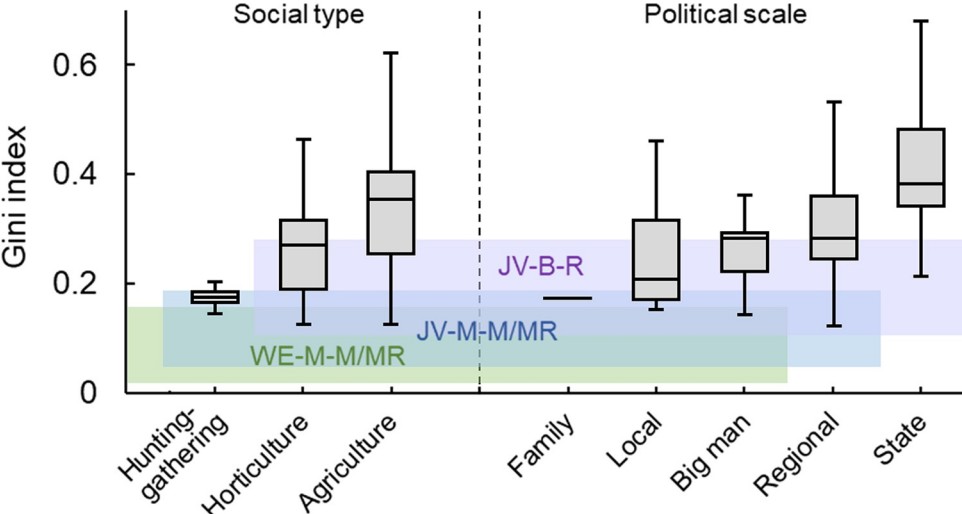

**Fig 9. Gini index in the Old and New Worlds in the post-Neolithic period.** (Kohler et al. 2017 [36]) Copyright Clearance Center's RightsLink® License Number: 5750640321076.

for household wealth and presents the results in a box-and-whisker graph. The Gini indices increase as the social type shifts from hunter-gathering to horticulture or agriculture, or as the political scale expands from family to big man, local, regional, or state. This is because the domestication of large mammals, the expansion of agricultural land, and the development of horse-mounted warfare promoted wealth accumulation and increased inequality.

Fig 9 shows the calculation results in Figs 7 and 8 with the WE-M-M/MR and JV-M-M/MR models for parameter variations in the light green and light blue bands, respectively. Additionally, the calculation result of the joint venture/redistribution model (JV-B-R) from reference [27] is shown in a light purple band. This model is the same as the JV-B model in Eq (1) and the redistribution without considering moral responsibility or risk vulnerability (specifically, in Eq (4), put $\rho_{Mi} = 1$) setting $t_p = 10^4$, $\xi = 0.5$, $\delta_w = 0.1$ and 0.2.

Fig 9 shows that the WE economy model is close to the social type of hunter-gathering and the political scale of family; the WE economy and joint venture/redistribution models are close to the minimum value among the various samples of social type and political scale; the joint venture/redistribution model, without considering moral responsibility and risk vulnerability, is close to the horticulture type and local/big man scales. The agriculture type and regional/ state scale, where wealth is increasingly accumulated and unevenly distributed, can be described as a combination of the power economy and the market economy described in the Introduction section. Fig 9 shows that reducing wealth inequality requires a transformation of the human economy based on moral responsibility and risk vulnerability. Thus, the restoration of a face-to-face economy where the social context is communicated.

## Discussion

This study modeled the WE economies and joint ventures by addressing moral responsibility and risk vulnerability, which are not addressed in the conventional econophysics models. The results for the two show that moral responsibility can be an economic instrument instead of inhuman money, and a comparison between the two shows that WE economies have a greater potential to reduce wealth inequality than joint ventures. In a joint venture, the profits and losses from the wealth contributed by each person are distributed according to their responsibilities, whereas in a WE economy, the profits and losses from the wealth contributed as a co-adventurer or community of destiny are distributed according to each other's moral responsibility ratio and risk vulnerability ratio. This study is the first to demonstrate the effectiveness of the mixobiotic society shown by philosopher Deguchi [12–14]; foundational communism shown by Graeber [8]; and advanced recovery of reciprocity shown by Karatani [9], inherited from Mauss's theory of gifts [37] and Kropotkin's theory of mutual aid [38], using an econo-physical approach.

The WE economy model is analogous to a hunter-gatherer society, as described in Fig 9. In this society, each tribal member brings food obtained through hunting and gathering to the communal square once and then distributes the food according to moral responsibility and risk vulnerability. By contrast, the joint-venture model is more akin to a horticultural or an agrarian society, as described in Fig 9. In this society, villagers cooperate in water management and seedling planting; however, food from the land belongs to the villagers who own the land, and the storage of food creates inequality. The redistribution model for joint ventures is similar to the role of the rich and chief in providing relief to the poor and the socially vulnerable based on moral responsibility. These are reminders of the combination of a joint economy *(mudaraba, murabaha,* and *salam)* and redistribution *(waqf, sadaqah,* and *zakat)* in the Islamic economy.

The effectiveness of wealth distribution and redistribution based on moral responsibility and risk vulnerability in both WE and joint venture economies suggests that both factors are

inherently important, and a shift from a money economy to a credit economy is possible. The money economy is based on equivalent exchanges divorced from the social context, whereas the credit economy is based on moral and mutual aid. Taking a long-term view of human history, Graeber notes that money and credit economies have alternated, for example, in the Middle Ages, Islamic societies transitioned from a money economy to a credit economy, and we are currently in a transition from a money economy to a credit economy [8]. The results of this study support Graeber's vision and encourage a transformation from a capitalist economy, which currently produces inequality, to a humanistic mutual-aid economy.

The resilience of the WE economy and joint ventures/redistribution based on moral responsibility for the distribution of wealth shows that it has the potential to break the vicious cycle of social unrest and stabilize society. The robustness of both models to parameter variation indicates that both have the generality to reduce inequality in diverse economic environments. Furthermore, the distribution of wealth, based on both moral responsibility and risk vulnerability, tilts toward more vulnerable people. This will contribute to a more inclusive society, for example, by helping children take the future, young people lead the next generation, and the socially vulnerable.

However, WE economies have the disadvantage of being more susceptible to free riders than joint ventures with redistribution. This can be viewed as a governance issue for a co-adventurer or community of destiny. Economist Ostrom cites the collective choice of operating rules, effective monitoring, and graduated sanctions as design principles for the commons [39]. Of these, there is a fear that monitoring and sanctions will lead to bad WE (totalitarianism, peer pressure), as indicated by Deguchi, but the collective choice of operating rules, including free riders or consensus building to recognize each other's moral responsibility, will be important.

In his book "The Moral Economy," economist Bowles presents the trilemma of Pareto efficiency, preference neutrality, and voluntary participation, citing limiting preference neutrality as a solution thereof [40]. The limitations of preference neutrality are the moral sensitization of free riders and the fostering of fellowship and cooperation; once these have been achieved, respect for both voluntary participation and the efficiency of the WE economy will be ensured, as indicated by Deguchi. Moreover, since societies and communities are multi-layered, an apparent free rider in one community can move to another based on voluntary participation.

Given the disadvantages of WE economies, even if compensated for by collective choice and restrictions on preference neutrality, such economies are suitable for relatively small-scale local societies and communities with common moral and social norms through face-to-face communication. WE economies require mutual recognition of moral responsibility and risk vulnerability. Specific examples include worker cooperatives [41], in which workers hold labor, investment, and management, and platform cooperatives [42, 43] and community wealth building [44], which are based on joint ownership and democratic decision-making by users and workers. To implement the WE economy through information technology, initiatives such as the Social Co-Operating System [45], which combines an operational loop that promotes cooperative behavior with a collegial loop that supports consensus building, would be useful. To expand the social scale of the WE economy, digital technologies that enable closer face-to-face communication and retention of social context are expected [46, 47].

In societies larger than WE economies, the redistribution of joint ventures based on moral responsibility and risk vulnerability is complementary and effective. In other words, it constitutes a multi-layered network of WE economies and joint ventures/redistribution and performs joint ventures/redistribution between the WE economies. However, redistribution should be based on norms and morals, as in the Islamic economy, rather than on hierarchies, as shown by Graeber [8], or plunder by power, as shown by Karatani [9]. The Islamic economy

is highly compatible with the WE economy as it emphasizes a real, face-to-face, and mutual-aid economy, and balances self-interest and altruism through various institutions [10, 11]. Just as Islamic societies triggered a shift to a credit economy in the Middle Ages [8], so too can the Islamic economy inspire a shift to a capitalist economic alternative today.

As a practical challenge, it is unrealistic to suddenly switch the current economy entirely to a WE economy based on moral responsibility and risk vulnerability. Just as Yunus started a social business called Grameen Bank in the beginning [48], the WE economy will start with small steps. For example, it could be a WE economy community of producers or consumers in the form of a worker coop or platform cooperative. It could also be a WE-economy circulation community, such as a supply chain, renewable energy, or local mobility, as a regional sphere [49] connecting rural and urban areas. Subsequently, once they are on track, a network of joint ventures/redistribution is established among the communities, gradually expanding the harmony of the WE economy. This will not be a top-down business transformation by governments and capitalists, but a bottom-up business transformation by the people toward what Graeber and Karatani call a human economy and Deguchi calls a mixbiotic society.

In this study, the simulations were conducted using moral responsibility and risk vulnerability on a straight line in a two-dimensional plane based on the literature [15]. However, the basic relationship and trends between WE economies and joint ventures/redistribution should not change, even if both are distributed on the plane. It is known that the mind perception changes with mental states [50, 51] and non-human objects [52, 53]. In the future, it is expected that artificially intelligent agents with morality are expected to emerge [14]. Although mental perceptions are expected to change depending on the contexts of economic actors and activities and social relations, the fundamental importance of the WE economy should remain the same.

This study modeled only the basic WE economy and joint ventures/redistribution; therefore, the absolute values of the parameters and calculation results do not necessarily reflect the real economy. However, as it is a basic model that discards details, it presents the effectiveness of moral responsibility and risk vulnerability, the complementary of the WE and joint economies, and the direction of economic transformation toward reducing wealth inequality. To apply Deguchi's WE economy in practice, focusing on community revitalization, as advocated by economist Rajan and policy scholar Hiroi, is key [54, 55]. Because the community is the third pillar to the state and the market. It will lead the current society, full of wealth inequality and social unrest, to a human mutual aid economy, and a fair and inclusive society. Note that this study assumes several constraints for econophysical modeling: there is mutual recognition of moral responsibility and risk vulnerability among economic agents and these mind perceptions are numerically available in economic activity. Future research should include analytical studies based on modeling and parameterization that reflect the real economy, psychological research on mind perception in economic activities, empirical research through case studies and fieldworks on economic activities based on moral responsibility and risk vulnerability, and social movements to spread the WE economy and transform the money economy into a credit economy.

## Supporting information

**S1 Fig. Character factor scores on two dimensions of mind perception.** (Gray, Gray, and Wegner 2007 [15]) Copyright Clearance Center's RightsLink® License Number: 5695221129353.
(TIF)

**S1 File. Programs and the resulting figures.**
(ZIP)

## Acknowledgments

This research was conducted as part of the "Toward Better 'Smart WE': From East Asian Humanities and Social Sciences to a Value Multi-Layered Society" project. I want to thank Professor Yasuo Deguchi of Kyoto University, the principal investigator of this project, whose ideas of "fundamental incapability," "Self as WE," and "mixbiotic society" directly motivated this study. I would also like to thank my colleagues at the Hitachi Kyoto University Laboratory of the Kyoto University Open Innovation Institute for their cooperation, and Editage (www.editage.jp) for English language editing.

## Author Contributions

**Conceptualization:** Takeshi Kato.

**Data curation:** Takeshi Kato.

**Formal analysis:** Takeshi Kato.

**Investigation:** Takeshi Kato.

**Methodology:** Takeshi Kato.

**Software:** Takeshi Kato.

**Validation:** Takeshi Kato.

**Visualization:** Takeshi Kato.

**Writing – original draft:** Takeshi Kato.

**Writing – review & editing:** Takeshi Kato.

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
