## [Decision Letter · Decision Letter 0]

20 Feb 2024

PONE-D-23-43602WE economy: Potential of mutual aid distribution based on moral responsibility and risk vulnerabilityPLOS ONE

Dear Dr. Kato,

Thank you for submitting your manuscript to PLOS ONE. After careful consideration, we feel that it has merit but does not fully meet PLOS ONE’s publication criteria as it currently stands. Therefore, we invite you to submit a revised version of the manuscript that addresses the points raised during the review process.

I recommend that it should be revised taking into account the changes requested by the reviewers. Since the requested changes include Minor Revision, I would like to give you a chance to revise your manuscript. However, the manuscript must undergo a next round of review by the same reviewer.

We look forward to receiving your revised manuscript.

Kind regards,

Baogui Xin, Ph.D.

Academic Editor

PLOS ONE

Journal Requirements:

"JSPS Topic-Setting Program to Advance Cutting-Edge Humanities and Social Sciences Research Grant Number JPJS00122679495"

4. Please expand the acronym “JSPS” (as indicated in your financial disclosure) so that it states the name of your funders in full.

"This research was conducted as part of the “Toward Better ‘Smart WE’: From East Asian Humanities and Social Sciences to a Value Multi-Layered Society” project (JSPS Topic-Setting Program to Advance Cutting-Edge Humanities and Social Sciences Research Grant Number JPJS00122679495). I want to thank Professor Yasuo Deguchi of Kyoto University, the principal investigator of this project, whose ideas of “fundamental incapability,” “Self as WE,” and “mixbiotic society” directly motivated this study. I would also like to thank my colleagues at the Hitachi Kyoto University Laboratory of the Kyoto University Open Innovation Institute for their cooperation. "

"JSPS Topic-Setting Program to Advance Cutting-Edge Humanities and Social Sciences Research Grant Number JPJS00122679495"

6. We note that your Data Availability Statement is currently as follows: [All relevant data are within the manuscript and its Supporting Information files.]

Reviewers' comments:

Reviewer's Responses to Questions

**Comments to the Author**

1. Is the manuscript technically sound, and do the data support the conclusions?

Reviewer #1: Yes

Reviewer #2: Yes

2. Has the statistical analysis been performed appropriately and rigorously? 

Reviewer #1: N/A

Reviewer #2: Yes

3. Have the authors made all data underlying the findings in their manuscript fully available?

Reviewer #1: Yes

Reviewer #2: Yes

4. Is the manuscript presented in an intelligible fashion and written in standard English?

Reviewer #1: Yes

Reviewer #2: Yes

5. Review Comments to the Author

Reviewer #1: The manuscript offers an innovative approach to wealth distribution through the introduction and analysis of three models. However, it could benefit from a more robust data and methodology section, ensuring the reliability of the simulation results and the representativeness of the data used. The empirical section requires further enrichment, possibly through the inclusion of diverse datasets, case studies, or field experiments to validate the models and assess their performance in various economic contexts. Addressing these aspects would significantly enhance the paper's rigour, practical applicability, and contribution to the field.

Reviewer #2: 1. P. 7, line 153: The notation used for risk vulnerability may be incorrect. Please review and confirm.

2. P. 9, lines 175-176: The subscript "k" is introduced without a clear definition. Please clarify what "k" represents.

6. PLOS authors have the option to publish the peer review history of their article (what does this mean?). If published, this will include your full peer review and any attached files.

Reviewer #1: No

Reviewer #2: No

---

## [Author Response · Author response to Decision Letter 0]

22 Mar 2024

Following the editor's instruction, I have submitted three items: 'Response to Reviewers,' 'Revised Manuscript with Track Changes,' and 'Manuscript' without tracked changes.

---

## [Editor Report · Decision Letter 1]

25 Mar 2024

WE economy: Potential of mutual aid distribution based on moral responsibility and risk vulnerability

PONE-D-23-43602R1

Dear Dr. Kato,

We’re pleased to inform you that your manuscript has been judged scientifically suitable for publication and will be formally accepted for publication once it meets all outstanding technical requirements.

Kind regards,

Baogui Xin, Ph.D.

Academic Editor

PLOS ONE
---

## [Editor Report · Acceptance letter]

3 May 2024

PONE-D-23-43602R1 

PLOS ONE

Dear Dr. Kato, 

I'm pleased to inform you that your manuscript has been deemed suitable for publication in PLOS ONE. Congratulations! Your manuscript is now being handed over to our production team.

Kind regards, 

on behalf of

Professor Baogui Xin 

Academic Editor

PLOS ONE